# Novel Coronavirus Infection (COVID-19) in Humans: A Scoping Review and Meta-Analysis

**DOI:** 10.3390/jcm9040941

**Published:** 2020-03-30

**Authors:** Israel Júnior Borges do Nascimento, Nensi Cacic, Hebatullah Mohamed Abdulazeem, Thilo Caspar von Groote, Umesh Jayarajah, Ishanka Weerasekara, Meisam Abdar Esfahani, Vinicius Tassoni Civile, Ana Marusic, Ana Jeroncic, Nelson Carvas Junior, Tina Poklepovic Pericic, Irena Zakarija-Grkovic, Silvana Mangeon Meirelles Guimarães, Nicola Luigi Bragazzi, Maria Bjorklund, Ahmad Sofi-Mahmudi, Mohammad Altujjar, Maoyi Tian, Diana Maria Cespedes Arcani, Dónal P. O’Mathúna, Milena Soriano Marcolino

**Affiliations:** 1University Hospital and School of Medicine, Universidade Federal de Minas Gerais, Belo Horizonte, Minas Gerais 30130-100, Brazil; israeljbn@ufmg.br (I.J.B.d.N.); smangeon@gmail.com (S.M.M.G.); milenamarc@gmail.com (M.S.M.); 2Cochrane Croatia, University of Split School of Medicine, Split 21000, Croatia; nensi.cacic@yahoo.com (N.C.); ana.marusic@mefst.hr (A.M.); ajeronci@mefst.hr (A.J.); tinapoklepovic@gmail.com (T.P.P.); izakarijagrkovic@yahoo.com (I.Z.-G.); 3Department of Sport and Health Science, Technische Universität München, 80333 Munich, Germany; dr.hebatullah.mohamed@gmail.com; 4Department of Anaesthesiology, Intensive Care and Pain Medicine, University of Münster, 48149 Münster, Germany; 5Department of Surgery, Faculty of Medicine, University of Colombo, Colombo 00700, Sri Lanka; umeshe.jaya@gmail.com; 6School of Health Sciences, Faculty of Health and Medicine, The University of Newcastle, Callaghan 2308, Australia; Ishanka.Weerasekara@uon.edu.au; 7Department of Physiotherapy, Faculty of Allied Health Sciences, University of Peradeniya, Peradeniya 20400, Sri Lanka; 8Cochrane Iran Associate Centre, National Institute for Medical Research Development, Tehran 16846, Iran; meysam.abdar@gmail.com (M.A.E.); ahmad.pub@gmail.com (A.S.-M.); 9Cochrane Brazil, Evidence-Based Health Program, Universidade Federal de São Paulo, São Paulo 04021-001, Brazil; vinicius_civile@yahoo.com.br; 10Cochrane Brazil, Universidade Paulista, São Paulo 04057-000, Brazil; nelson.carvas96@gmail.com; 11Laboratory for Industrial and Applied Mathematics (LIAM), Department of Mathematics and Statistics, York University, Toronto, ON M3J 1P3, Canada; robertobragazzi@gmail.com; 12Faculty of Medicine, Lund University, SE-221-00 Lund, Sweden; maria.bjorklund@med.lu.se; 13Department of Internal Medicine, University of Toledo, Toledo, OH 43606, USA; tujjarmd@hotmail.com; 14The George Institute for Global Health, University of New South Wales, Sydney, New South Wales 2052, Australia; mtian@georgeinstitute.org.cn; 15The George Institute for Global Health, Peking University Health Science Center, Beijing 100088, China; 16Department of Cardiovascular and Thoracic Surgery, Zhongnan Hospital, Wuhan University, Wuhan 430070, China; maria-d18@hotmail.com; 17Helene Fuld Health Trust National Institute for Evidence-based Practice in Nursing and Healthcare, College of Nursing, The Ohio State University, Columbus, OH 43210, USA; omathuna.6@osu.edu; 18School of Nursing, Psychotherapy and Community Health, Dublin City University, D04V1W8 Dublin, Ireland

**Keywords:** novel coronavirus, SARS-CoV-2, COVID-19, scoping review, meta-analysis

## Abstract

A growing body of literature on the 2019 novel coronavirus (SARS-CoV-2) is becoming available, but a synthesis of available data has not been conducted. We performed a scoping review of currently available clinical, epidemiological, laboratory, and chest imaging data related to the SARS-CoV-2 infection. We searched MEDLINE, Cochrane CENTRAL, EMBASE, Scopus and LILACS from 01 January 2019 to 24 February 2020. Study selection, data extraction and risk of bias assessment were performed by two independent reviewers. Qualitative synthesis and meta-analysis were conducted using the clinical and laboratory data, and random-effects models were applied to estimate pooled results. A total of 61 studies were included (59,254 patients). The most common disease-related symptoms were fever (82%, 95% confidence interval (CI) 56%–99%; *n* = 4410), cough (61%, 95% CI 39%–81%; *n* = 3985), muscle aches and/or fatigue (36%, 95% CI 18%–55%; *n* = 3778), dyspnea (26%, 95% CI 12%–41%; *n* = 3700), headache in 12% (95% CI 4%–23%, *n* = 3598 patients), sore throat in 10% (95% CI 5%–17%, *n* = 1387) and gastrointestinal symptoms in 9% (95% CI 3%–17%, *n* = 1744). Laboratory findings were described in a lower number of patients and revealed lymphopenia (0.93 × 10^9^/L, 95% CI 0.83–1.03 × 10^9^/L, *n* = 464) and abnormal C-reactive protein (33.72 mg/dL, 95% CI 21.54–45.91 mg/dL; *n* = 1637). Radiological findings varied, but mostly described ground-glass opacities and consolidation. Data on treatment options were limited. All-cause mortality was 0.3% (95% CI 0.0%–1.0%; *n* = 53,631). Epidemiological studies showed that mortality was higher in males and elderly patients. The majority of reported clinical symptoms and laboratory findings related to SARS-CoV-2 infection are non-specific. Clinical suspicion, accompanied by a relevant epidemiological history, should be followed by early imaging and virological assay.

## 1. Introduction

In December 2019, a series of cases of a novel virus causing respiratory infections in humans was observed in patients after they had visited a local market in the Chinese city of Wuhan [1]. The novel virus was named “2019 novel coronavirus (2019-nCoV/SARS-CoV-2)” and was first isolated on 7 January 2020. Since then, the virus has spread worldwide and has infected 167,515 patients globally, causing 6606 deaths as of 16 March 2020 [2,3]. Patients infected with the virus may either be asymptomatic or may experience mild to severe clinical symptoms such as pneumonia, respiratory failure and death [4]. The syndrome of clinical symptoms caused by SARS-CoV-2 is called “coronavirus disease” (COVID-19) [5].

The SARS-CoV-2 is an enveloped, single-stranded RNA virus that can be transmitted from human to human [6,7]. Bats have been identified as a key reservoir of coronavirus in China [8,9]. The SARS-CoV-2 is about 50% genetically identical to MERS-CoV and about 79% identical to SARS-CoV, to which it has a similar receptor-binding domain structure [10].

Due to the novelty of the virus and the short duration of the SARS-CoV-2 outbreak, only a limited and scattered body of scientific evidence is available on various aspects of COVID-19. The first systematic review on the topic was published in February 2020; however, it lacked a defined search strategy and public and transparent protocol, and it included only eight epidemiological or clinical cohort studies, providing a narrow focus on clinical symptoms only [11].

We therefore aim to analyze the published scientific literature on the SARS-CoV-2 infection worldwide concerning the clinical, epidemiological, laboratory and radiological characteristics of COVID-19, as well as its course, severity, and treatment options.

## 2. Experimental Section

This scoping review follows the Meta-analysis of Observational Studies in Epidemiology (MOOSE) guidelines and is reported in accordance with the Extended Preferred Reporting Items for Systematic Reviews and Meta-Analyses Statement for Scoping Reviews (PRISMA-ScR) [12]. The review protocol was submitted to PROSPERO (CRD42020170623) and published on the Open Science Framework (OSF) (Appendix A).

### 2.1. Literature Search and Selection Criteria

MEDLINE, CENTRAL, EMBASE, Scopus and LILACS databases were searched for eligible publications from 01 January 2019 to 24 February 2020. The search strategy (Appendix A) was designed and conducted in collaboration with an information specialist based in Sweden. Publications regarding SARS-CoV-2 were eligible for inclusion, regardless of study design and publication language. Therefore, case reports, case series, correspondences and editorials were processed in order to identify patient data. A confirmed case of SARS-CoV-2 was defined and mostly diagnosed using the triple algorithm (epidemiological history, clinical symptoms and laboratory or radiological findings) as a standard procedure proposed by the World Health Organization. Studies involving animal experimentation were excluded. Reference lists of relevant studies were screened to identify any missing publications. All searches and title and abstract screenings, as well as study selection, were performed independently by two investigators. Discrepancies were resolved by consensus. Articles deemed potentially eligible were retrieved for full-text review. Non-English publications were translated by a native/fluent speaker. Ethics approval was not necessary.

### 2.2. Outcomes

The primary outcomes were all-cause mortality rate and clinical symptoms. Other outcomes comprised demographic characteristics, co-morbidities, incubation period, laboratory results, radiological and computer tomographic findings, types of treatment provided (oxygen supplementation or various ventilation therapies), admission to the intensive care unit (ICU), days in ICU and length of hospital stay. We processed data from baseline to follow-up. If a study reported multiple follow-ups, the most recent data were included.

### 2.3. Data Extraction and Quality Assessment

Data extraction and risk of bias assessment were performed independently by two investigators. Discrepancies were resolved by consensus. Data for patients were analyzed individually to avoid overlap. If any overlapping was suspected, corresponding authors were contacted to clarify the discrepancy. Additionally, we performed an assessment comparing information from the hospital that the patients were admitted to and the epidemiological week in order to avoid overlap. Two researchers independently assessed the risk of bias of selected studies using the Methodological Quality and Synthesis of Case Series and Case Reports Protocol proposed by Murad et al. [13], derived from the Newcastle–Ottawa Scale (NOS), except for two questions not relevant to our scoping review (“Was there a challenge/re-challenge phenomenon?” and “Was there a dose–response effect?” [14]). Disagreements were resolved by consensus. It is important to mention that only clinical symptoms, mortality and laboratory findings were included in the meta-analysis performed.

### 2.4. Statistical Analysis

We extracted data for the number of events and total patients to perform proportion meta-analysis using R software, with the “meta” package (version 4.9–6), the “metaprop” function for proportion data and the “metamean” function for continuous data. For studies that presented continuous data as medians and inter-quartile ranges, the estimate of the means and standard deviations was performed according to the method described by Wan et al. [15].

We conducted a meta-analysis using the clinical and laboratory data. We presented pooled results of proportion with their respective 95% confidence intervals (CI) by the inverse variance method with a random-effects model, using the DerSimonian–Laird estimator for τ^2^. We adjusted data by Freeman–Tukey double arcsine transformation and confidence intervals were calculated by the Clopper–Pearson method for individual studies. For continuous data, we presented pooled results of means with their respective 95% CI by the inverse variance method with a random-effects model, using the DerSimonian–Laird estimator for τ^2^. In this case, we adjusted the data using the untransformed (raw) means method. Heterogeneity was assessed by Cochran’s Q test considering a statistically significant value for *p* < 0.1 and Higgins I^2^.

Subgroup analyses were performed to assess whether there was a difference in the results for clinical variables and mortality, with respect to patient backgrounds (China vs. other countries).

## 3. Results

Our search retrieved 2701 records, of which 236 were duplicates. We shortlisted 426 publications which met the inclusion criteria for full-text analysis (Figure 1) and identified 66 records reporting clinical data. Seven additional relevant studies were identified from the references of included studies (Figure 1).

Of the 73 records, 13 were excluded as overlaps or duplicate publications. Thus, 60 studies were included in this review [6,16,17,18,19,20,21,22,23,24,25,26,27,28,29,30,31,32,33,34,35,36,37,38,39,40,41,42,43,44,45,46,47,48,49,50,51,52,53,54,55,56,57,58,59,60,61,62,63,64,65,66,67,68,69,70,71,72,73,74,75,76,77]. We included three studies even though they investigated patients from the same sample because different parameters were analyzed in the three studies (Chen L et al. [16], Feng K et al. [17], Tang N et al. [18]). The main publication languages were English and Chinese, with one study in Korean.

### 3.1. Study and Patient Characteristics

The main characteristics of the included studies are summarized in Appendix A. Characteristics of excluded studies are summarized in Appendix A. There were 20 case reports, 37 case series and 3 epidemiological reports, with a total of 59,254 patients from 11 different countries. Overall, the male/female ratio was 1.08 and the age of the population ranged from 3 months to 99 years. The most prevalent co-morbidities were hypertension, diabetes, chronic liver disease and smoking.

### 3.2. Risk of Bias

The quality assessment of each study is summarized in Appendix A. Risk of bias was generally high due to the study design of case reports or case series. Therefore, the certainty of the evidence was very low for all studies included.

### 3.3. Clinical Symptoms

Forest plots for clinical symptoms are shown in the Appendix A. Hereafter, we present the incidence of symptoms, confidence interval (CI) and the number of patients providing data for meta-analysis (*n*). One of the included studies reported only mortality data and no clinical symptoms. As it included data for over 40,000 patients, the *n* for each clinical symptom is much lower than the total number of patients. The most common symptom was fever (82%, 95% CI 56%–99%, *n* = 4410), then cough, with or without sputum, was reported in 61% (95% CI 39%–81%, *n* = 3985) of cases, muscle aches and/or fatigue in 36% (95% CI 18%–55%, *n* = 3778), dyspnea in 26% (95% CI 12%–41%, *n* = 3700), headache in 12% (95% CI 4%–23%, *n* = 3598 patients), sore throat in 10% (95% CI 5%–17%, *n* = 1387 and gastrointestinal symptoms in 9% (95% CI 3%–17%, *n* = 1744) of patients.

### 3.4. Chest Imaging Findings

Chest imaging findings were described in detail in the majority of included studies (*n* = 51). Among patients who underwent chest radiologic examination, the most common abnormalities were opacities (bilateral or unilateral, with or without pleural effusion, *n* = 22 patients), multiple ground-glass opacities (*n* = 20 patients) and infiltrate (unspecific to lobe involvement, *n* = 4 patients). Only six patients showed normal chest radiographical findings. With regards to computer tomography, prevailing findings were ground-glass opacities (accompanied or not by septal thickening, *n* = 1204 patients), infiltration abnormalities (*n* = 9 patients) and parenchymal consolidation (*n* = 325 patients). Normal CT results were present only in 8 patients. Chest imaging in SARS-CoV-2 pneumonia seems to be similar to ordinary viral pneumonia, with some particularities. Patchy ground-glass shadow, which is more commonly peripheral/sub-pleural, with irregular shape and distribution of alveolar opacification and without geometric blurred vessels, were described. A single lung (single or multiple lobes) or both lungs (without a rigid pattern) may be affected. Ground-glass opacity nodules, which may progress to a larger opacity or irregular alveolar consolidation, were other common tomographic abnormalities reported. In these cases, there is an infected secretion in the pulmonary alveolus, with blurred vessels, which defines a more severe evolution of the disease [19].

### 3.5. Laboratory Findings

Of the sixty included studies that reported laboratory findings, 56 (*n* = 58,663 patients) reported the confirmation of the novel coronavirus infection using real-time PCR. One study (*n* = 2 patients) used genetic analysis. Three studies (*n* = 529 patients) did not report the confirmation method. Two studies (*n* = 2 patients) reported positive assay in asymptomatic patients. Other laboratory findings are presented in Table 1.

In our meta-analysis for C-reactive protein, one particular study (Lin X et al.) had a low number of patients (2). For this reason, while performing our statistical analysis, we perceived a negative level of this biomarker due to the limitation of our estimator software, which could not calculate or consider the sample size appropriately. However, after a sensitivity analysis had been carried out, we still observed a trend of elevated CRP among the studies selected-sensitivity analysis for CRP: MRAW (untransformed means) = 38.15 (95% CI 29.36–46.95, I^2^ = 64%). Few studies assessed hemoglobin level, eosinophils and monocyte count, coagulation profile, serum glucose level or serum amyloid A protein, so they are not presented given the limited amount of data.

### 3.6. Management and Mortality

Pharmacological and/or supportive interventions were reported in 26 publications (1876 patients). In six reports, only summary information (prescription or not) was described, with no specific information about the medication dose or route of administration. Antivirals were provided to 815 patients, with the most commonly used agents being oseltamivir (66.8%, *n* = 544 patients), arbidol (6.6%, *n* = 54 patients), ganciclovir (9.3%, *n* = 76 patients), and ritonavir (17.3%, *n* = 141 patients). Overall, 815 patients received antivirals. Antibiotics were used in 836 patients, but most of the studies did not mention the exact compound administered or indication for the use of antibiotics. Single patients received different antibiotics (vancomycin, azithromycin, meropenem, cefaclor, cefepime and tazobactam), 73 patients were administered linezolid and 3 patients received moxifloxacin. Other medications used were corticosteroids (*n* = 183 patients), alpha-interferon (*n* = 19 patients), immunoglobulin (*n* = 232 patients) and antifungal drugs (*n* = 47 patients). It was not possible to perform subgroup analysis to check the effectiveness of antivirals, antibiotics and other medications on the prognosis. Studies were descriptive in nature and lacked suitable control groups for comparison of clinical efficacy. This was because there was considerable heterogeneity in the reporting of therapeutic agents used. Therefore, the methodological discrepancies and heterogeneity in the reporting precluded this analysis.

Although studies did not provide details on pO_2_ or SpO_2_, they reported that in patients requiring supportive therapy, 38.9% received supplementary oxygen through a nasal cannula, 7.1% required non-invasive ventilation, 28.7% required mechanical ventilation and 0.9% required extracorporeal membrane oxygenation (ECMO). Other supportive treatments were fluid therapy, vitamin K1, continuous renal replacement therapy and blood transfusions. Information on the type of supportive treatment provided was not specified in 11.2% of cases. Overall, 8.3% (140 out of 1686 patients) required intensive care treatment. Due to the lack of data we were not able to assess the length of ICU stay or hospitalization. All-cause mortality assessment is shown in Table 2.

### 3.7. Epidemiological Findings

Epidemiological data on SARS-CoV-19 were reported in three studies from China that included a total of 54,498 patients, of which 53,991 (99.0%) were confirmed cases [20,21,22,23]. The majority of cases were from the Hubei province (75.8%), most them from Wuhan. The majority of patients described were of working age (20–60 years (66.7%) [21]) and a higher incidence of infection was seen in males (0.31 vs. 0.27 per 100,000 population) [23]. Median time from onset of disease to diagnosis was 5 (interquartile ratio 2–9) days [20]. The median incubation period ranged between 4.5 and 4.7 days [22,23]. Most cases were described as mild (81.4%), 13.9% were severe and 4.7% were critical [21]. The majority of fatalities were in patients ≥ 60 years-old (81.0%) [21]. Yang et al. estimated a case fatality rate (CFR) of 3.06% (95% CI 2.02%–4.59%) in their cohort [23]. Male sex, age ≥ 60 years, delay in diagnosis and diagnosis of severe pneumonia were associated with increased CFR [23]. In China, the outbreak risk increased until the 23rd of January and decreased thereafter [20]. However, the cumulative number of diagnosed cases and fatality is still rising [21].

## 4. Discussion

This is the first report to provide a comprehensive overview of the available evidence on the SARS-CoV-19 outbreak. Sixty studies were included (case reports, case series or epidemiological reports) with a total of 59,254 patients from 11 countries. The most common symptoms in patients with SARS-CoV-19 infection were fever (82%), cough (61%), muscle aches and/or fatigue (36%) and dyspnea (26%). The most common chest radiographic abnormalities reported were bilateral opacities, multiple ground-glass shadows, infiltrate shadows and consolidation in the lungs, and thickening of the pulmonary texture. The most frequent computed tomographic abnormalities were ground-glass opacities, septal thickening and parenchymal consolidation. Mortality among the patients infected with SARS-CoV-19 was 3.0% and most of the data were from China. In epidemiological studies from China, male sex, age ≥ 60 years, delay in diagnosis and diagnosis of severe pneumonia were associated with increased mortality rates.

The symptoms of COVID-19 are not specific, which makes it clinically indistinguishable from other viral respiratory illnesses. Although fever was the most common manifestation of COVID-19, the fever-free period of infection remains unknown, which may cause patients not to be identified initially, and some patients may even be asymptomatic. Non-respiratory symptoms such as headache, fatigue, sore throat and gastrointestinal symptoms should not be overlooked, and high suspicion should be maintained in those with a positive epidemiological history, followed up by thorough clinical evaluation by a healthcare provider.

The radiological abnormalities found in patients with SARS-CoV-19 pneumonia were similar to those found in other types of viral pneumonia [78]. However, for SARS-CoV-19, the chest tomography pattern of ground-glass and consolidative pulmonary opacities, often with a bilateral and peripheral lung distribution, is emerging as the CT hallmark of COVID-19 infection [24]. Artificial intelligence (AI) has been recently raised as a potential tool to enhance care, and there are several studies suggesting that AI can perform as well as or better than humans in imaging analysis for the diagnosis of different diseases [79]. A China-based technology company has developed an image-reading system that uses AI to detect abnormalities of possible coronavirus pneumonia. As AI can read a CT scan in seconds, it can help to assist physicians making fast judgments [80]. Furthermore, as PCR-based diagnosis requires long time periods until results are available, CT imaging with AI could serve as a surrogate for physicians when a quick decision is necessary [80]. The AI solution was launched on 19 February 2020 and by 28 February 2020 it had already been used on scans for 5000 patients [81]. However, due to the lack of evidence regarding the use of AI interventions, we recommend that more studies should be performed. Until then, the findings from these interventions must be used with caution to avoid incorrect decision-making.

Due to low specificity, laboratory tests may not be useful in establishing the diagnosis of COVID-19, however they can help appraise the clinical condition of a patient and may be indicative of COVID-19, resulting in further testing with PCR and radiological studies.

The rate of patients requiring admission to an intensive care unit was relatively low (8.3% among 1686 patients in which this outcome was assessed) but still may cause significant burden for healthcare systems worldwide. The use of supplementary oxygen therapy (38.9%), non-invasive (7.1%) and invasive ventilation (28.7%) and even ECMO (0.9%) was surprisingly high among the 1876 patients in which any kind of pharmacological and/or supportive intervention was reported, but no parameters of hypoxia, such as pO_2_ or SpO_2_, or even respiratory rate, were provided. Therefore, the real disease severity cannot be known in those cases and it cannot be known whether supplemental oxygen was used as a therapeutic or preventive measure. Recent studies have documented the remaining conflicting aspect regarding non-invasive ventilation in ill patients with SARS-CoV-19. For this reason, we suggest that physicians may not utilize non-invasive ventilation during clinical management, especially those with acute respiratory disease syndrome, due to the lack of evidence; if a patient’s status does not get better or even get worse, mechanical ventilation can be favored [82,83]. So far, no specific treatment or vaccination for COVID-19 is available. More rigorously designed trials and investigations will be necessary to better understand the role of antiviral drugs.

Asymptomatic cases, patients who had mild manifestations of disease and those who were not tested for SARS-CoV-19 infection might have been missed, resulting in a higher mortality rate. Epidemiological studies testing groups of both asymptomatic and symptomatic individuals may prove helpful in exploring this hypothesis. On the other hand, some fatal cases, especially in patients with multiple and advanced co-morbidities, might have led to death prior to seeking medical attention.

In the epidemiological studies, the majority of fatal cases were reported in the older age group; therefore, these patients require early diagnosis, followed by intensive monitoring and appropriate therapy. Since all the epidemiological studies included in our review were limited to China, there is a need for reports from other countries in order to obtain a global perspective on the epidemic. The rise in the incidence of diagnosed cases worldwide will hopefully provide an incentive for other countries to record and share their epidemiological data.

The finding of a lower mortality rate outside of China is limited by the small sample size, but it is in line with WHO data. A number of theories have been created regarding this epidemiological observation and they deserve specific attention. Firstly, genomic mutation of the SARS-CoV-19 during its spread around the world may play a role in this process [84]. Secondly, different aspects in the quality of medical treatment in Wuhan medical facilities may account for this difference in mortality rates. Ji Y et al. [85] reported that a comprehensive analysis of the Chinese Center for Disease Control and Prevention data showed clear disparities in mortality rates between Wuhan (>3%) and other provinces of China (around 0.7%). The authors hypothesized that this is likely to be related to the quick rise in the number of infections in the epicenter.

This scoping review has limitations. With new data being published on a daily basis, this review can only provide results up to 24 February 2020. However, we believe that new publications do not modify the trend and the main characteristics found for the disease. Due to the novelty of the virus and the short timeframe since the beginning of the outbreak, the certainty of the evidence is limited, given that most evidence currently is available as case reports and case series. Nevertheless, given the lack of higher quality studies, inferences from such reports can be helpful in guiding decision-making [86]. Finally, there was considerable heterogeneity in the data, especially for the clinical symptoms, which we interpret not to be due not to major publication bias, but rather as a result of the small sample sizes in studies published so far (with a small number of patients experiencing less common symptoms), or because of the heterogeneity of the disease itself.

## 5. Conclusions

Further research on all aspects of the disease is needed to better understand the infection, especially in regard to the rate of asymptomatic patients and beneficial treatments. Systematic analyses like this review will be needed as new clinical data are reported. Prospectively designed observational and clinical trials will help improve the certainty of the available evidence. The data on SARS-CoV-2 should continue to be shared transparently and promptly, and a global repository may help with this.

## Figures and Tables

**Figure 1 jcm-09-00941-f001:**
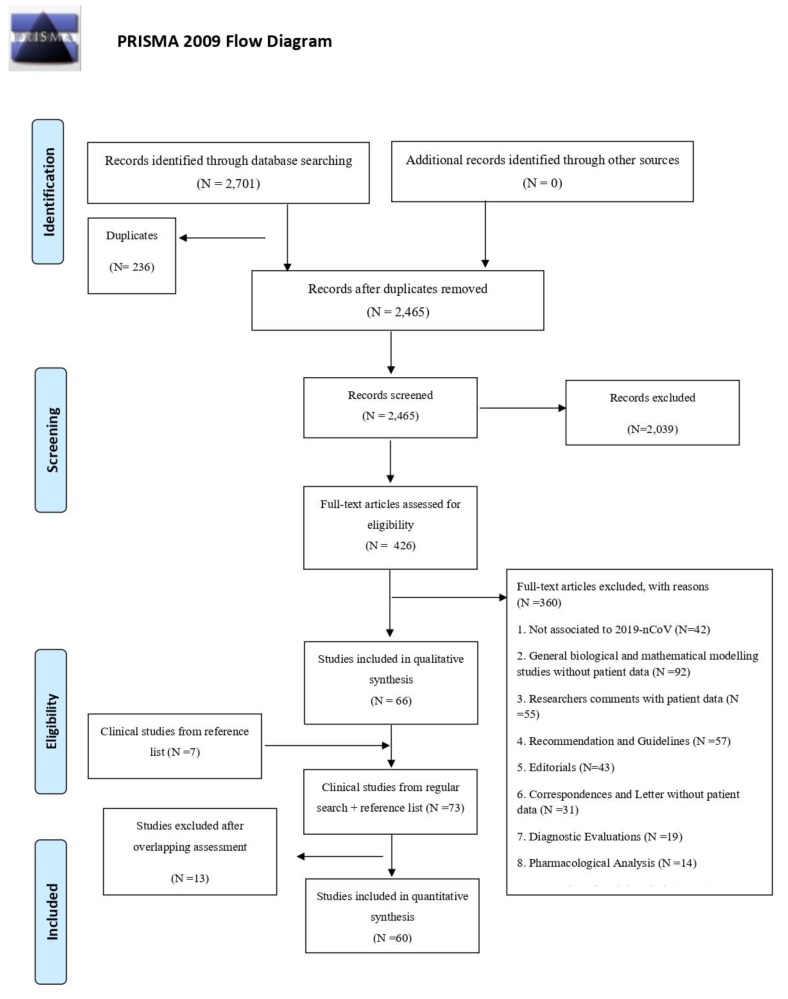
Prisma flow diagram.

**Table 1 jcm-09-00941-t001:** Laboratory findings in patients infected with SARS-CoV-2.

Laboratory Test	No. of Studies	Total Patient No.	Values in Physiologic Range *n* (%)	Values > Physiologic Range *n* (%)	Values < Physiologic Range *n* (%)	Lost to Follow up *n* (%)
**Inflammatory markers**						
**C-RP**	25	1637	427 (26.1%)	900 (55.0%)	-	310 (18.9%)
**ESR**	7	105	NA	88 (83.8%)	-	NA
**PCT**	12	1463	NA	98 (6.7%)	NA	NA
**IL-6**	1	99	NA	51 (52.0%)	NA	NA
**Peripheral blood profile**						
**Total WBC**	32	1747	1109 (63.5%)	155 (8.9%)	469 (26.8%)	14 (0.8%)
**Neutrophils**	20	204	143 (70.1%)	48 (23.5%)	6 (2.9%)	7 (3.4%)
**Lymphocytes**	25	464	159 (34.3%)	47 (10.3%)	256 (55.2%)	2 (0.4%)
**Platelets**	11	218	NA	64 (29.4%)	25 (11.5%)	NA
**Blood biochemistry**						
**ALT**	12	1316	NA	211 (16.0%)	NA	NA
**AST**	18	1420	NA	254 (17.9%)	NA	NA
**LDH**	11	283	NA	157 (55.5%)	NA	NA
**D-dimer**	16	1573	NA	527 (33.5%)	NA	NA

Abbreviations: C-RP = c-reactive protein, ESR = erythrocyte sedimentation rate, PCT = procalcitonin, IL-6 = interleukin-6, WBC = white blood cell count, ALT = alanine transaminase, AST = aspartate-transaminase, LDH = lactate dehydrogenase, URL = upper reference limit, LRL = lower reference limit.

**Table 2 jcm-09-00941-t002:** Summary of findings (SOF) table for all-cause mortality.

Outcome	Study Population	Incidence (95%CI)	Higgins I^2^-Test	Certainty of the Evidence (GRADE)
**All patients**	31 studies (53,631 patients)	**0.3 (0.0–1.0)**	**83%**	(+) very low
**Chinese patients**	28 studies (5632 patients)	**0.5 (0.0–1.4)**	**85%**	(+) very low
**Patients from other countries**	3 studies (41 patients)	**0.0 (0.0–1.4)**	**0%**	(+) very low

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
