# Peer review of "Novel Coronavirus Infection (COVID-19) in Humans: A Scoping Review and Meta-Analysis"

_jcm, 2020, doi:10.3390/jcm9040941_

Round 1

Reviewer 1 Report

Thank you for the high level of timeliness research. Some revisions seem to be needed. 1. Please amend all of "2019-nCoV" to " SARS-CoV-2" in the manuscript. 2. In order to make accessibility of this research, please add COVID-19 or other term in the title. 3. line 41: In this study, only clinical and laboratory data were included in the meta-anlaysis. However, line 41 can confuse the reader. Thus, It is considered the fact should be addressed in the absctract. 4. 2.3 Data Extraction and Quality Assessment: Please address that only clinical and laboratory data were included in the meta-anlaysis. 5. line 113: "Statistical analysis" is considered to be subheadings of chapter 2.3? 6. lines 147-148: "Risk of bias" is considered to be subheadings of chapter 3.1? 7. lines 152-157: Please describe what the "n" means exactly. number of researches? number of patients? If the "n"s mean the number of patients included in the meta-analysis, the "n"s are different from the Supplementary appendix 3. 8. Supplementary appendix 3: 7.1 Please order the plots according to the sequence in the manuscript(lines 152-157). 7.2 In order to make readability higher, please rearrange the "Country = China" above the "Country = Out of China" in figure 1, 2, and 7 7.4 Please add the number of patients in each research in figure 8, 9, and 10. 7.3 In figure 10, the CI interval of Lin, X. et al is under the zero, which is not logical. 9. lines 159, 164,165: It seems that each "n" means the number of studies and the number of patients, respectively. In order to make clear, please address this. 10. line 175: The number of researchers included in this study was sixty six. Sixty means the number of studies that included the laboratory findings such as C-RP, ESR and PCT? 11. lines 191-192: In order to reduce the confusion, please address that "n" means the number of patients. 12. line 198: This is the one of the questions that a lot of clinician and researcher want to know. Please add the reasons. 13. lines 245- 254: AI can be the one of the tools that can help to contain the COVID-19.However, researches have showed that AI needs to be more calibrated because of its sensitivity on the algorithm and the version of the algorithm, which means AI has substantially high risk for wrong decision. The control of the emerging infectious disease should highly depend on the scientific evidences because of uncertainty.The controlling should highly based on the social distancing, quarantine and laboratory test. As the authors know, in order to make more evidence, this research is needed. To summarize, AI can be one of the options. However, it needs more evidences. I understand the author's opinion. It needs to address this issue with more cautions.

Author Response

Dear reviewer, 

In attachment our responses to your valuable comments.

Reviewer 2 Report

JCM jcm-751181

“Novel Coronavirus Infection in Humans: a Scoping Review and Meta-Analysis”

The authors present a review of available clinical data and mortality data on COVID19. These types of reviews are useful for synthesizing available knowledge. The forest plots and meta-analyses of reported symptoms seem particularly valuable. Nonetheless, I have concerns about the quality of the paper given the fact that I detected a number of errors. I recognize that mistakes occur when preparing something quickly, as the circumstances necessitate, but I have one particular comment that strikes me as poor logic. Note that, as a statistician, I am focusing my review on the quantitative analyses of the paper.

Major:

  1. In the abstract and in several other places, the authors refer to 61 studies with 101,905 patients, but this is misleading to readers. It implies that we have data from over 100,000 patients, but in fact, these numbers are driven by two large studies (40,000+ patients) of the same population in China. It is only meaningful to add populations that you know have no overlap, which is all of China plus any other countries reporting data. Ignoring the redundancy of these studies gives their data incredible influence over some of the aggregated statistics, such as male to female ratio. It concerns me that the authors would report this sample size as they have done, and makes me wonder about the extent of overlap in any of the cohorts summarized in the symptom meta-analyses.
  2. Many mistakes of varying importance:
    1. In Table S1, information for Zhu et al. appears to be missing.
    2. There seem to be a number of inconsistencies between the sample sizes reported in the text and in the forest plots. For example, fever is based on 2452 observed patients, yet the text reports 6856 (large discrepancy). Cough is based on 2027, yet the text reports 2410. In fact, none of them seem to match exactly.
    3. Introduction, line 71. Reference 11 includes 10 studies. Also, it was published in the Journal of Medical Virology, not the Lancet. https://onlinelibrary.wiley.com/doi/full/10.1002/jmv.25735
    4. Inconsistency between papers used in the analysis and papers cited. See Yang Y et al 2020 analysis of 4021 patients laboratory confirmed cases. This is included in the forest plot, but not listed among the references. On line 218, results from this same analysis are attributed to the wrong paper, as reference [23] is not Yang Y et al 2020. (Disclosure: I am an author on this paper.)
  3. Section 2.1, line 89. Why is it necessary to have an epidemiological history to be a confirmed case if you have laboratory or radiological findings? Particularly laboratory findings. Is there any concern that this will exclude laboratory-confirmed cases with no obvious exposure history?
  4. In Figures 8 and 9, why are studies listed that contribute no data (weight 0%)? If this is because they do not report a measure variance, this should be explained more clearly in either the statistical section or the results section. They might be better placed in a separate part of the figure since they do not contribute to the aggregate analysis.
  5. Discussion of interpretation of the case fatality ratio should include a clearer description of the challenges of estimating this quantity with data in an emerging outbreak

Minor:

  1. Abstract and Introduction: Update virus and disease name. First paragraph of the introduction needs to define SARS-CoV-2.
  2. Abstract: Lymphopenia 109/L?
  3. Introduction: When reporting case counts (e.g. Line 60), report the date when these figures were reported rather than “more than.” “As of XX date, ….”
  4. Introduction, line 72. The use of the world trials is inappropriate here. These are epidemiological studies or clinical cohorts.
  5. Section 2.3, line 113. Statistical analysis should be a new section.
  6. Section 2.3: R package documentation should be referenced.
  7. Table S3. I would suggest dropping the columns that are NA only and reiterating their omission in the table caption.

Author Response

Dear reviewer, 

In attachment our most sincere answers for your honorable comments. Thank you so much!

Round 2

Reviewer 2 Report

The authors have adequately addressed my comments. I have no further questions.